# Investigating the Trackability of Silicon Microprobes in High-Speed Surface Measurements

**DOI:** 10.3390/s21051557

**Published:** 2021-02-24

**Authors:** Min Xu, Zhi Li, Michael Fahrbach, Erwin Peiner, Uwe Brand

**Affiliations:** 1Physikalisch-Technische Bundesanstalt (PTB), Bundesallee 100, 38116 Braunschweig, Germany; Zhi.li@ptb.de (Z.L.); uwe.brand@ptb.de (U.B.); 2Institute of Semiconductor Technology (IHT), Technische Universität Braunschweig, Hans-Sommer-Straße 66, 38106 Braunschweig, Germany; m.fahrbach@tu-braunschweig.de (M.F.); e.peiner@tu-bs.de (E.P.); 3Laboratory for Emerging Nanometrology (LENA), Langer Kamp 6 a/b, 38106 Braunschweig, Germany

**Keywords:** roughness measurement, piezoresistive microprobe, high-speed surface measurement

## Abstract

High-speed tactile roughness measurements set high demand on the trackability of the stylus probe. Because of the features of low mass, low probing force, and high signal linearity, the piezoresistive silicon microprobe is a hopeful candidate for high-speed roughness measurements. This paper investigates the trackability of these microprobes through building a theoretical dynamic model, measuring their resonant response, and performing tip-flight experiments on surfaces with sharp variations. Two microprobes are investigated and compared: one with an integrated silicon tip and one with a diamond tip glued to the end of the cantilever. The result indicates that the microprobe with the silicon tip has high trackability for measurements up to traverse speeds of 10 mm/s, while the resonant response of the microprobe with diamond tip needs to be improved for the application in high-speed topography measurements.

## 1. Introduction

Surface roughness plays a great role in diverse fields, such as semiconductor technology, automotive manufacturing, and medicine engineering. It is an important predictor of the performance of a mechanical component [1,2]. The international standard ISO 4287/4288 [3,4] specifies various parameters, *R_a_, R_q_, R_z_, R_sk_*, etc., for the evaluation of the surface roughness. One of the most widely used parameters, the arithmetical mean deviation of the surface height *R_a_* is expressed mathematically by
(1)$$R_{a}=\frac{1}{N}\overset{N}{\underset{i=1}{\sum }}\left|Z_{i}\right|$$
where *N* is the number of measured points in a sampling length, and *Z_i_* is the ordinate values of the roughness profile.

The measurement of surface roughness demands a technology with high accuracy, high throughput, relatively large measurement range, and low probing force. The measurement methods of surface texture and roughness can be divided into two categories: contact stylus instruments and optical instruments such as vertical scanning interferometry (VSI). Although the optical methods have the advantages of high throughput and no probing force, their utilization is limited by the optical properties and surface structures of the artifacts. The optical instruments have difficulties in measuring the surfaces with slopes. Undesirable light reflection and diffraction effects decrease signal quality. A study performed by Jaturunruangsri [5] proves that the stylus method is more accurate than the VSI instrument in the roughness measurements for hard materials.

However, one drawback of the contact stylus instruments lies in the low throughput. During the measurement, the stylus scans line by line. It is especially time-consuming for large range measurements. The measurement throughput of the contact stylus instrument depends on the traverse speed of the motion stages during the measurement. The maximum traverse speeds of the state-of-the-art stylus instruments are in the range of 1–3 mm/s [6,7]. The work of Arvithe Davinci et al. [7] indicates that the traverse speed of the stylus profilometer affects the roughness measurement results significantly. When the speed is below 500 µm/s, the roughness measurement results are stable. If the speed is further increased, a sharp variation in the results happens. This points out that an improvement of the trackability of the stylus at high speed is necessary.

As to the research on more rapid stylus instruments, Morrison developed a prototype stylus profilometer that can measure with speeds up to 5 mm/s in 1995 [8,9]. Since then, there was very little progress on developing stylus instruments with higher traverse speeds. One obstacle to further improving the traverse speed of the stylus lies in the fact that the stylus probe will lose contact to the surface and thus profile fidelity as it moves over steep features. No loss of tracking and signal fidelity at high traverse speeds above 10 mm/s is a hard requirement for stylus probes.

The Physikalisch-Technische Bundesanstalt (PTB) developed a slender piezoresistive silicon cantilever type microprobe together with the Institute for Semiconductor Technology of Technical University of Braunschweig and the Forschungsinstitut für Mikrosensorik GmbH (CiS) Erfurt [10,11,12,13,14,15,16]. Cantilevers of 5 mm length, 200 µm width, 50 µm thickness, and a mass of about 0.1 mg, as shown in Figure 1 were used, which are commercially available as CAN50-2-5 from the CiS GmbH (https://www.cismst.de/en/loesungen/mikrotastspitzen/ (accessed on 4 January 2021)). At the free end of the microprobe is an integrated probing tip with a height of more than 100 µm, a cone angle of 40°, and a radius of about 0.1 µm. The long and sharp tip enables the microprobe for the roughness measurement on the surface with *R_a_* value under 25 µm [17].

When a force acts vertically on the probing tip, the cantilever is deflected and a full bridge piezoresistive strain gauge on the cantilever close to its clamping measures the bending of the cantilever. The microprobe converts the deflection into a voltage output. The nonlinearity of the conversion influences the measurement accuracy directly. The nonlinearity between the output voltage variation and the deflection is about 0.3%. The deflection range of the microprobe is up to 200 µm. It means that the error caused by the conversion nonlinearity is less than 0.6 µm in measuring a height of 200 µm.

The signal fidelity of the contact stylus probe is influenced by many factors, such as:The geometrical structure of the probe tip, such as the opening angle and the tip radius. Inappropriate tip geometry leads to contact positions other than the tip end and results in measurement deviations from the artifact surface. This is called tip-sample convolution effect [18].The non-linearity between the probe output signal and the surface variation.The dynamic behavior of the probe. The probe loses track as it traverses over steep features if the dynamics of the probe are not high enough.The measurement bandwidth of the probe [19], usually defined by the first free resonant frequency of the probe.

Among the above factors, the demand on the last two factors, the dynamics and the measurement bandwidth of the probe, increases with the traverse speed. Hence these two factors become especially important in high-speed measurements. Compared to the existing conventional styli (with masses of several mg, the first free resonant frequency in the order of hundreds of Hz, and a cone angle of either 60° or 90°), the microprobe has a much lower mass, higher resonant frequency, and a sharper tip. It is suggested that the microprobe has a high potential for high-speed measurements.

This paper demonstrates the microprobe as a promising stylus probe candidate for high-speed roughness measurement at 10 mm/s. It is intended to give an uncomplicated method to evaluate the trackability of the microprobe. The analysis results can be proved with simple and feasible experiments, and the theoretical analysis and experimental results will indicate the improvement direction of the microprobe for better performance.

In the following, a theoretical model is proposed to investigate the dynamics of the microprobe. The steepest feature that the microprobe can measure without loss of tracking is examined, the resonant frequencies of the microprobes are analyzed and measured, and proof-of-principle experiments were performed and detailed.

## 2. Modelling the Dynamic Behavior of the Microprobe

Since the limited tracking fidelity is noticed first at steep features, we investigate the behavior of the microprobe on artifact surfaces with sharp variations.

The steepest feature that a probe can measure is restricted by the trackability, the tip form, and the mounting angle of the probe. In this section, we focus on the influence of the dynamic behavior of the microprobe.

It should be noted that the tip works as a mechanical low pass Gaussian filter and smoothens the sharp features in contact measurements [20]. If the cantilever tracks all the frequency components passing through the low pass filter formed by the tip, it can be considered that the cantilever can track the surface with fidelity. In other words, the dynamics of the cantilever are high enough for the measurement.

The microprobe is supposed to traverse across a falling edge and tip trajectory is drawn. The steepest feature on the surface should vary slower than the tip trajectory, otherwise, the tracking will be lost.

### 2.1. Theoretical Analysis

It is assumed that the artifact is an ideal step artifact with 90° inclined sidewalls and a height *H*, as presented in Figure 2. The falling edge of the top surface is set to be position 0, the origin of the x-z coordinate system.

The microprobe is mounted with tilt angle *θ.*

The tip traverses across the artifact surface and moves over the edge of the top surface (position 0). It loses contact with the surface and the deflection of the cantilever decreases. In the case the cantilever doesn’t touch the step artifact ground, i.e., under free-flight condition there is
(2)$$m\ddot{z}+d\dot{z}+kz=F_{z}\left(0\right)=\;F\left(0\right)\cos\theta$$
where *z* is the vertical deflection variation of the cantilever, F(0) is the deflection force on the cantilever at position 0, *F_z_* (0) is the vertical component of F(0), *k* is the spring constant of the cantilever, and *d* is its damping factor. In this calculation it is assumed that the whole mass of the cantilever is concentrated in the tip and *m* is the effective mass.

The reduced effective mass of the beam *m_b_ =* 0.24*M*, *M* is the total mass of the beam [21]. The effective mass of the whole cantilever *m* is the sum of the tip mass *m_t_* and the reduced effective mass of the beam *m_b_*:(3)$$m=\;m_{t}+m_{b}=\;m_{t}+0.24M$$

The vertical speed $\dot{z}\;$and the deflection variation *z* can be calculated by single and double integration, respectively, of the acceleration:(4)$$\dot{z}=\;\dot{z}\left(0\right)+\int_{0}^{t}\ddot{z}\;dt$$
(5)$$z=\;\int_{0}^{t}\dot{z}\;dt$$
where *t* is the time it takes to move the tip from position 0 to position *z*, $\dot{z}\;\left(0\right)$ is the vertical speed of the microprobe cantilever tip at position 0. 

As the top surface of the step feature is flat, the initial condition exists:(6)$$\dot{z}\left(0\right)=0$$

Based on Equations (2)–(6), the trajectory of the cantilever tip under free-flight condition drawing the deflection variation *z* at the time *t* is only determined by the initial probing force F(0) for a microprobe with the given tilt angle *θ*.

The angle *β*, the inclination of the steepest slope that the microprobe can track, is calculated through analysing the gradient of the tip trajectory ∇z(x) under free-flight condition:(7)$$\beta ={\;\tan}^{-1}\left(\nabla z\left(x\right)\right)$$
where *x* is the displacement along the *x* axis. 

In the case with constant traverse speed *v_x_*, there is
(8)$$\nabla z\left(x\right)=\;\frac{\dot{z}}{v_{x}}$$

The steepest slope feature is determined by the ratio of the vertical speed to traverse speed. With the increment of the traverse speed *v_x_*, the feature that the tip can track becomes less inclined.

The above analysis calculates the steepest slope feature that the microprobe can track. If the slope is known with angle *γ* and the trackability of the microprobe on the slope is evaluated, the effect of friction should be considered. The friction *F_f_* is calculated by:(9)$$F_{f}=\;\mu \left(F\left(0\right)\cos\theta -kz-d\dot{z}\right)\cos\left(\theta +\gamma \right)/\cos\theta$$
where *μ* is the coefficient of friction between the microprobe tip and the artifact surface.

### 2.2. The Effect of Tip-Sample Convolution

The vertical speed of the tip at the origin is 0, the microprobe tip at this position cannot track any slope according to Equation (8). However, it does not necessarily mean that the microprobe will lose track at this position since tip-sample convolution effect should be taken into consideration.

When the tip traverses across an arbitrary surface, the imaged profile is the convolution result of the surface with the tip shape, and sharp features are smoothed. The artifact surface after tip-sample convolution *S’* is calculated by [22]
(10)$$S^{\prime }=S⊕TIP$$
where *S* is the original artefact surface, and *TIP* is the tip surface.

It is defined that *S = s(x)*, *TIP = tip(u)*, *u_0_ ≤ u ≤ u_1_*, and *S’= s´(x)*. *u*_0_ and *u*_1_ are the lower and upper limits of abscissa values of the tip outline separately. There is
(11)$$s^{\prime }\left(x\right)=s\left(x\right)-\underset{u_{0}\leq u\leq u_{1}}{\min}\left(tip\left(u\right)-s\left(x+u\right)\right)$$

When the artifact surface other than the sampling position touches the tip, or in other words, when the tip surface other than the tip end touches the artifact surface, the tip-sample convolution occurs. 

If the inclination of the tip trajectory under free-flight condition is larger than that of an artefact surface after tip-sample convolution, we expect that the tip can track the surface with fidelity.

### 2.3. The Dynamic Behavior of the Microprobe with a Silicon Tip

For the microprobe with parameters listed in Table 1, the vertical cantilever tip movement versus time calculated using Equation (5) are drawn in Figure 3a with four different initial probing forces 32 µN, 64 µN, 96 µN, and 130 µN. It indicates that the larger the initial probing force is, the faster the tip drops and the larger the measurement range is. It costs 0.1 milliseconds for the tip dropping from *z* = 0 µm to *z* = −2 µm if the initial probing force is 32 µN. The time is reduced to 0.04 milliseconds if the initial probing force is increased to 130 µN.

In the following, the trackability of the microprobe is investigated with respect to step features with 30° and 70° inclined sidewalls.

First, the tip-sample convolution effect is simulated as shown in Figure 3b. The surfaces are convoluted according to Equation (11) by the tip with the radius of 0.1 µm, the nominal radius of the microprobe silicon tip. 

The tip trajectories under free-flight condition at the traverse speed of 20 µm/s, 10 mm/s, and 50 mm/s, calculated using Equation (5) are drawn in Figure 3c. The initial probing force is 96 µN. The slopes with 30° and 70° inclinations after tip convolution are also depicted for comparison. At the traverse speed of 10 mm/s, the tip trajectory falls more rapidly than both slopes, which means that the probe can track both slopes with signal fidelity. When the traverse speed is further increased to 50 mm/s, the tip trajectory falls slower even than the convoluted 30° slope at the falling edge and the probe loses track, which means the microprobe has no capability to track slopes >30° at a traverse speed of 50 mm/s, applying a probing force of 96 µN.

The steepest slope that the microprobe can track, only considering the dynamic behavior, is presented in Figure 3d. The slope inclination increases rapidly from 0° at the start position. Then it keeps relatively constant and begins to decrease when the cantilever comes close to its deflection-free position. Since tip convolution smoothens sharp features, the moderate gradient at the starting point of the tip trajectory does not restrict mapping the surface feature if an appropriate initial probing force is selected. These curves help us to select the appropriate probing force if rough information about the surface is available. For example, with a traverse speed of 10 mm/s, an initial probing force of 96 µN is needed to track slope features with a height of 9 µm and an inclination of 70°. The initial probing force can be decreased to 32 µN for features of 3 µm height and an inclination of 30°.

From the above analysis, it can be concluded that the 5 mm long silicon microprobe has high dynamics and can track steep features with inclination variations up to 70° at traverse speeds up to 10 mm/s.

### 2.4. The dynamic Behavior of the Microprobe with a Diamond Tip

Since undesired silicon tip abrasion exists in measurements, a diamond tip with a tip radius of 2 µm was glued to the microprobe to replace the silicon tip, as defined in DIN EN ISO 4288 [4]. The half opening angle of the diamond tip is about 45°, as shown in Figure 4. Limited by the half opening angle, the inclination of the slope feature that can be measured by the microprobe with the diamond tip is smaller than 45°.

With a height of about 210 µm and the base area of about 5 × $10^{6}\;µm^{2}$, the mass of the conical diamond tip is about 0.012 mg. The effective mass of the cantilever is increased to 0.036 mg (from 0.024 mg for the cantilever with silicon tip) and the dynamic performance of the microprobe is thus decreased. However, the surface variation after tip-sample convolution is moderated stronger by the 2 µm-radius tip than the 0.1 µm-radius silicon tip since the “cut-off” wavelength *λ_t_* of the low-pass Gaussian filter formed by the 2 µm-radius diamond tip is longer according to
(12)$$\lambda_{t}=2\pi \sqrt{AR}$$
where *R* is the tip radius and *A* is the amplitude of the surface feature.

In consequence, the microprobe with the diamond tip can track the 45° slope at a traverse speed of 10 mm/s with an initial probing force of 96 µN, as calculated using Equation (5) and shown in Figure 5.

**Figure 5 sensors-21-01557-f005:**
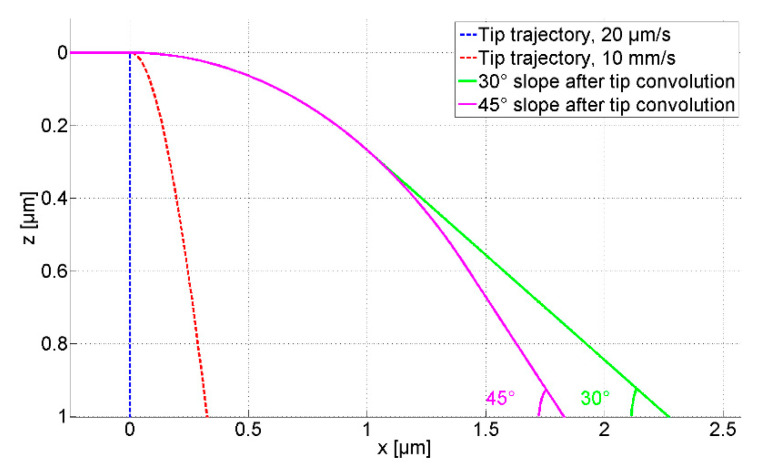
The trajectory of the microprobe with a diamond tip at different traverse speeds, with an initial probing force of 96 µN.

**Table 1 sensors-21-01557-t001:** Nominal parameters of the microprobe used to calculate the cantilever tip trajectories in Figure 3 and Figure 5.

Nominal Parameters	Symbol	Value
Length	*L*	5 mm
Width	*w*	200 µm
Thickness	*b*	50 µm
cross-section area	*A = wb*	$10^{-8}\;m^{2}$
Density	ρ	2330 kg/m^3^
Mass of the cantilever	M=ρLwb	0.1 mg
Mass of the tip	$m_{t}$	0.15 µg for the silicon tip of the CAN50-2-50.012 mg for the diamond tip
Half opening angle of the tip		20° for the silicon tip of the CAN50-2-5 45° for the diamond tip
effective mass of the cantilever	*m*	0.024 mg for the CAN50-2-5 0.036 mg for the cantilever with a diamond tip
Spring constant	*k*	8.45 N/m
Damping factor	*d*	0.001
Young’s modulus	*E*	169 GPa
Area moment of inertia	$I_{a}=\frac{wb^{3}}{12}$	2.093 ×$\;10^{-18}\;m^{4}$
Tilt angle	θ	15°

## 3. Frequency Response of the Microprobes

The frequency response of a probe is important since it is a decisive factor in the measurement bandwidth of the probe. The demand on the measurement bandwidth of the probe increases linearly with the traverse speed because the frequency *f* exerted to the tip by the surface structure increases with the traverse speed *v_x_* as given by
(13)$$f=\frac{v_{x}}{\lambda }$$
where *λ* is the spatial wavelength of the structure.

Not only the frequency response of the microprobe without contact to a surface, but more important the frequency response of the probe in contact to a surface should be known.

Neglecting the mass of the probing tip the fundamental resonant frequency of a cantilever with an approximately rectangular shape and without a tip at its free end [23] can be calculated by
(14)$$f_{0}=\;\frac{1.758}{\pi L^{2}}\sqrt{\frac{EI_{a}}{\rho A}}$$

Using the parameters listed in Table 1, the resonant frequency of the CAN50-2-5 microprobe is calculated to be f_0_ = 2.8 kHz. The frequency of the microprobe with an integrated silicon tip used in the measurement was measured to be about *f_0_* = 3.2 kHz, slightly higher than its nominal value, which is assigned to deviations from the nominal dimensions due to fabrication tolerances of the CAN50-2-5. Due to the effect of the added tip mass, a lower resonant frequency of *f_0_* = 2.2 kHz was measured with the microprobe with the diamond tip.

The contact resonant frequency of a probe is influenced by many factors, such as the tip radius, the artifact material, and the probing force. The first contact resonant frequency *f_c_* of a probe can be roughly estimated to be about three to five times of its first free resonant frequency *f_0_*.

If the “cut-off” spatial wavelength is *λ_t_* = 1 µm, the contact resonant frequency of the probe is demanded to be above f_c_ = 10 kHz for the traverse speed of *v_x_* = 10 mm/s according to Equation (13). This is just a rough estimation and not a hard criterion. The microprobe with a silicon tip (*f_c_* = 9.6 to 16 kHz) may meet this requirement while the microprobe with a diamond tip (*f_c_* = 6.6 to 11 kHz) possibly does not.

## 4. Experiment Results

### 4.1. “Tip Flight” Test

“Tip flight” tests (see Figure 6) were performed on flat smooth surfaces of a step artifact to prove the trackability of the microprobes. The idea was to measure with the microprobes across sloped features with different traverse speeds. To characterize tip flight, a normalized flight width was defined, which is the ratio of the flight width to the step structure height. It was used as a measure to evaluate the dynamics of the microprobes in comparison with the modeling described in Section 2 quantitatively.

The test was carried out with the instrument Profilscanner, a self-developed profiler at PTB [24,25,26]. The microprobe is glued and bonded on a microprobe holder and mounted on the tip scanning head of the Profilscanner which consists of an XYZ piezo stage (PI, model P-628.2CD for XY axes and P-622.ZCD for Z-axis) with a motion range of 800 µm × 800 µm × 250 µm (X × Y × Z).

Whether the XY stage can move steadily at the defined traverse speed influences the test results. The moving speeds of the XY piezo stage were measured before the ‘flight test’ with laser interferometers and the result proves that the XY piezo stage moves steadily at a speed up to 10 mm/s.

### 4.2. Artifacts and the Test Results

Two artifacts were used in the test. Artifact A is diamond turned from Cu and then coated with 15 µm Ni and 5 µm Cr on top (see Figure 7a). The nominal height is 10 µm and the slope inclination is about 30°. The nominal height of artifact B (see Figure 7b) is also 10 µm, but the slope inclination is 90°. It is made of steel. Because of the manufacturing tolerances, the edges of the top surface of artifact B are somewhat protruding.

Figure 8 shows the measured profiles by the CAN50-2-5 microprobe with the integrated silicon tip traversing across the artifacts with speeds of 20 µm/s, 5 mm/s, and 10 mm/s respectively. The initial probing forces are 96 µN for artifact A and 130 µN for artifact B. The measured heights of both artifacts significantly deviate from the nominal values given by the manufacturer. Since the height of artifact B is about 14 µm the static probing force needed to be increased to end up with the tip in contact with the ground surface of the artifact. Because of the influence of the mounting tilt angle (θ = 15°) and the half opening angle of the tip (20°), the measured slope angle of the artifact B is about 55° instead of 90°. For both sloped artifacts, A and B, the profiles acquired at different traverse speeds agree well and flight widths are zero. This result proves the high fidelity of the measured profiles related to the fast-responding dynamics of the microprobe with a silicon tip.

The tip flight test was repeated with the microprobe with a diamond tip. Since the mounting tilt angle (θ = 15°) and the half opening angle of the diamond tip (45°) leads to that the inclination that the diamond tip can measure is no more than 30°, only artifact A was measured with the microprobe with a diamond tip. Again, an initial static probing force of 96 µN was used. As shown in Figure 9, the tip flight widths at different traverse speeds again were zero. This indicates the dynamics of the microprobe with the diamond tip is good enough for measuring the slope of 30° inclination up to traverse speeds of 10 mm/s, confirming the theoretical model developed above. However, the tip vibrated during the measurements on the artifact surface at the higher speeds of 5 mm/s and 10 mm/s. It means that the resonant response of the microprobe with the diamond tip is not high enough for such high-speed measurements. The bandwidth of the microprobe with the diamond tip will have to be improved by increasing the stiffness or decreasing the mass of the tip.

The “tip flight” test results demonstrate that the dynamics of both microprobes are high enough for the measurements at the speed of 10 mm/s, but the bandwidth of the microprobe with a diamond tip should be raised to meet the demand of trackability.

In the next step of our work, we will decrease the length of the microprobe to increase the resonance frequency and investigate the trackability further.

## 5. Summary

To investigate the trackability of piezoresistive silicon microprobes for high-speed surface roughness measurements, a theoretical dynamic model was derived to examine the dynamics of the microprobes. The resonant response of the microprobes was analyzed and tip-flight tests were performed. Both the theoretical analysis and the experimental results prove that the microprobes with integrated silicon tip have the capability of tracking surfaces with high fidelity up to traverse speeds of 10 mm/s. However, the resonant response of a microprobe with a glued diamond tip of 90° opening angle will have to be improved to fulfill the demands of high-speed roughness measurements.

## Figures and Tables

**Figure 1 sensors-21-01557-f001:**
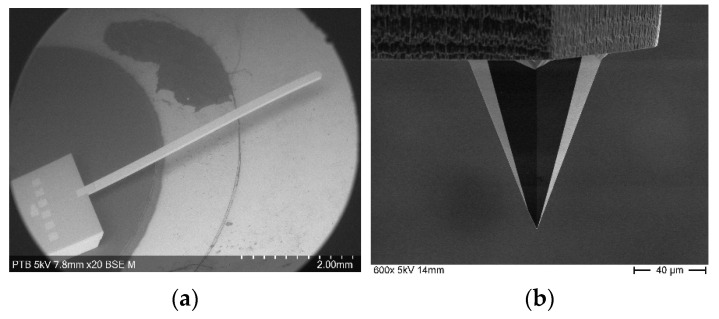
The scanning electron microscope images of the microprobe. (**a**) Sample of 5-mm long piezoresistive microprobe; (**b**) integrated silicon tip.

**Figure 2 sensors-21-01557-f002:**
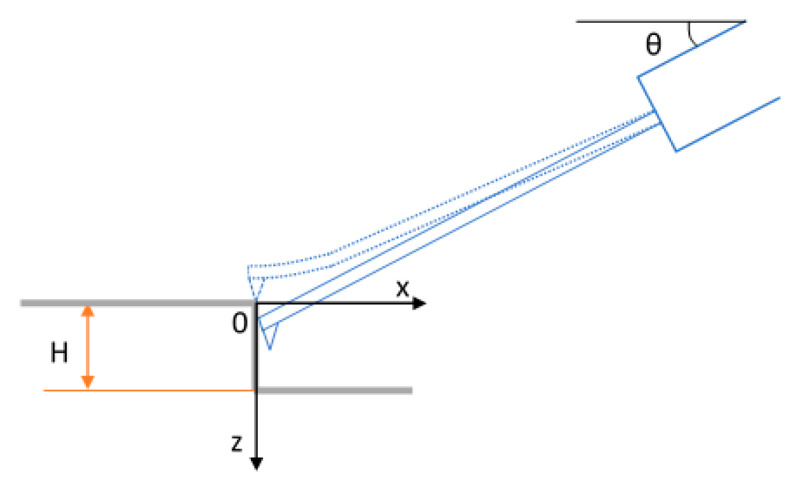
The deflection of the microprobe changes after passing the starting point of the falling edge (position 0) of the step artefact.

**Figure 3 sensors-21-01557-f003:**
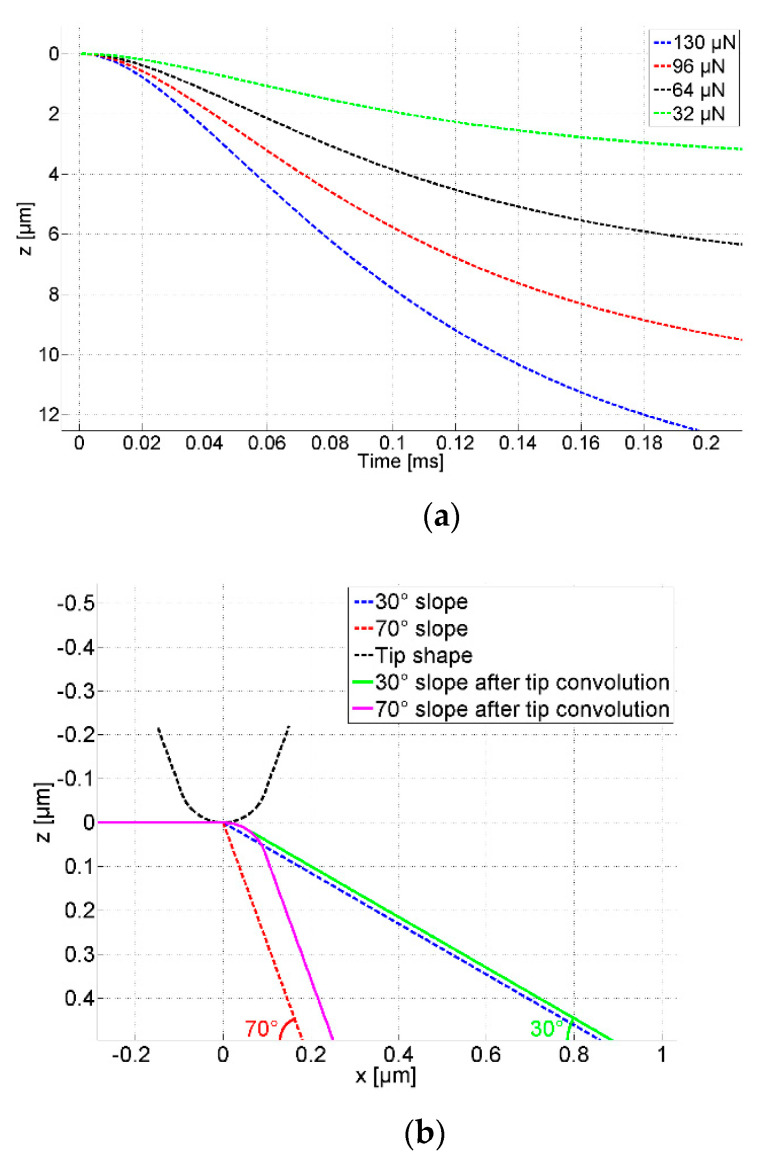
The dynamic behavior of the microprobe with a silicon tip. (**a**) The trajectory of the microprobe tip of the CAN50-2-5 drawing z with time under the condition of different initial probing forces; (**b**) The slope features before and after the tip convolution; (**c**) comparison of the tip trajectories and the slope features; (**d**) the steepest slope inclinations that the microprobe can track at the traverse speed of 10 mm/s.

**Figure 4 sensors-21-01557-f004:**
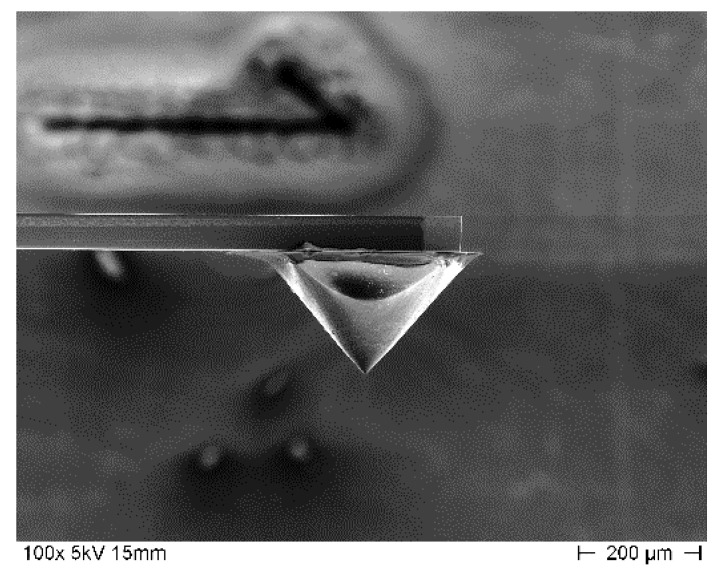
The microprobe with a glued 210 µm high diamond tip.

**Figure 6 sensors-21-01557-f006:**
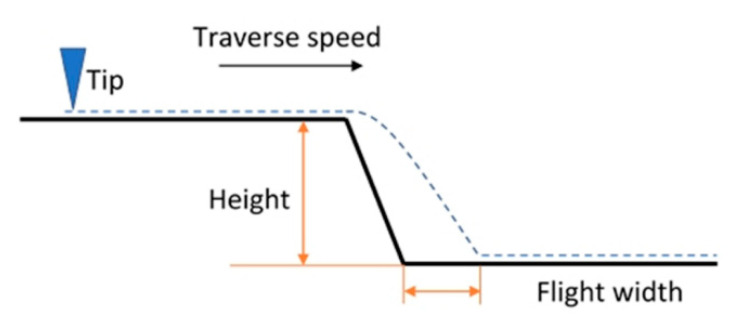
The tip flight test to measure the dynamic behavior of the microprobe.

**Figure 7 sensors-21-01557-f007:**
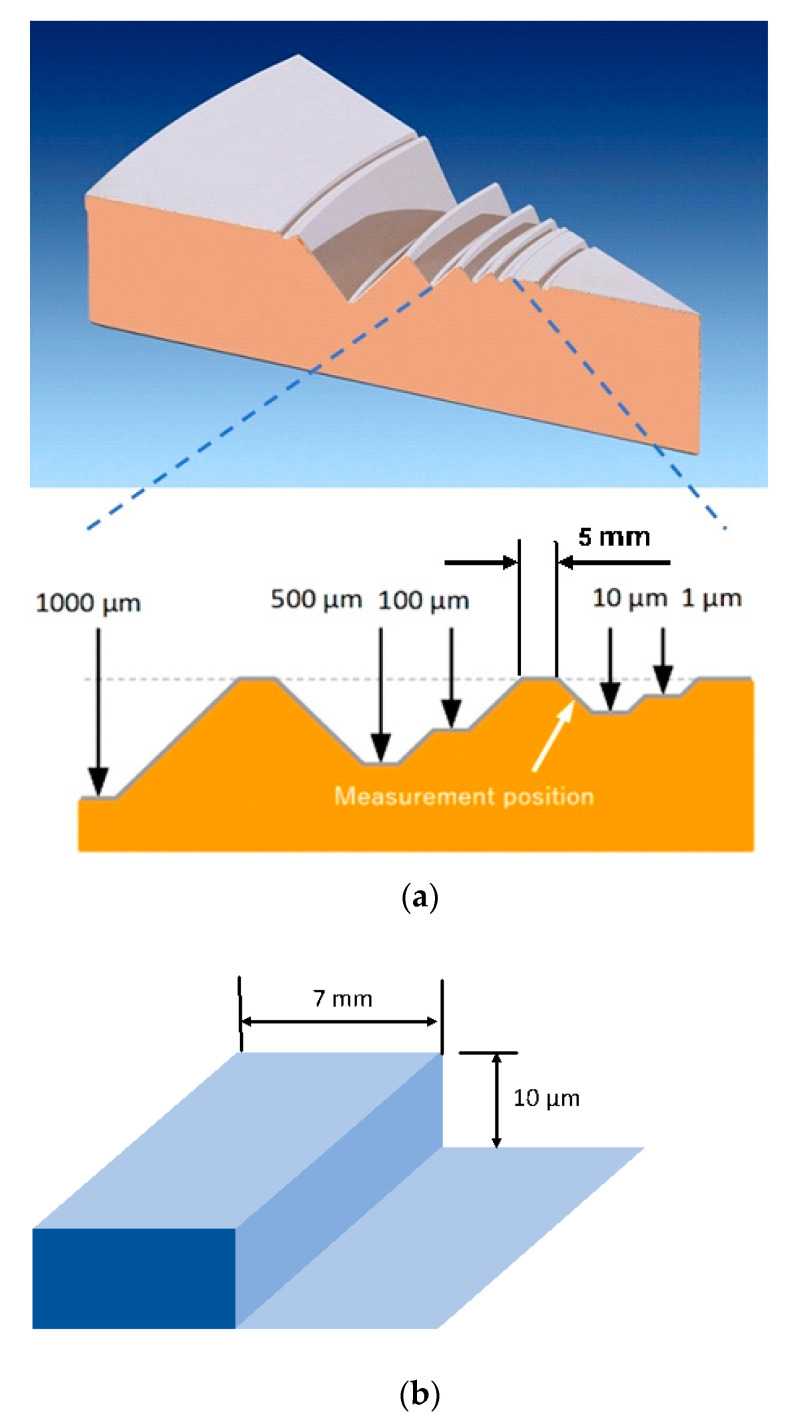
Artifacts used in ”tip flight” test. (**a**) artifact A; (**b**) artifact B. The artifacts aren’t drawn to scale.

**Figure 8 sensors-21-01557-f008:**
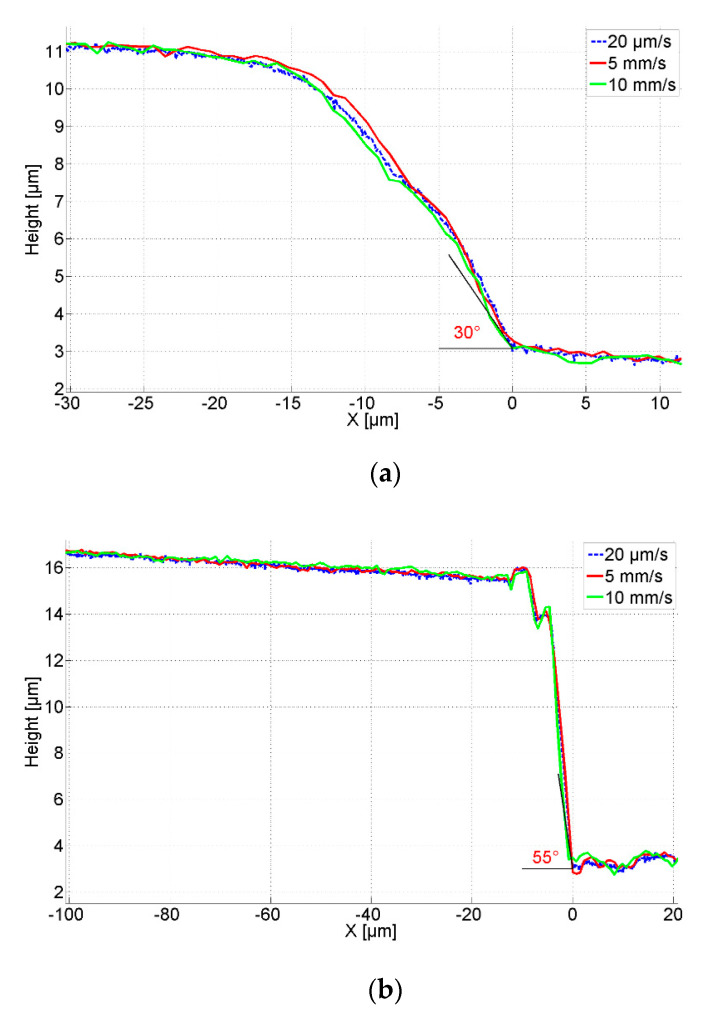
Measured profiles with the CAN50-2-5 microprobe with integrated silicon tip. (**a**) Measured profiles on artifact A (30° inclined sidewall) with an initial static probing force of 96 µN, at different traverse speeds; (**b**) Measured profiles on artifact B (90° inclined sidewall) with an initial static probing force of 130 µN, at different traverse speeds.

**Figure 9 sensors-21-01557-f009:**
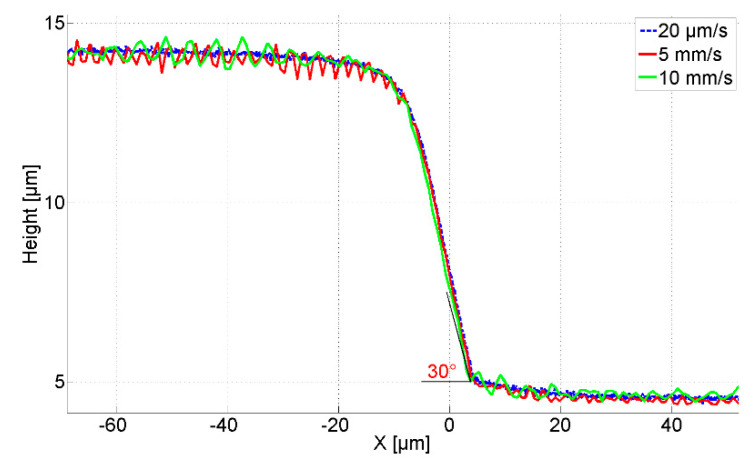
Measured profiles on artifact A using the microprobe with diamond tip with an initial static probing force of 96 µN, at different traverse speeds.

## Data Availability

All data and code will be made available on request to the correspondent author’s email with appropriate justification.

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
