# Peer review of "Investigating the Trackability of Silicon Microprobes in High-Speed Surface Measurements"

_sensors, 2021, doi:10.3390/s21051557_

Round 1

Reviewer 1 Report

The manuscript "Investigating the trackability of silicon microprobes in high
speed surface measurements" reports on theoretical and experimental studies on the the trackability of silicon microprobes in high speed surface measurements.

From a general point of view, the topic of the manuscript is interesting and worth of investigation. Generally, the manuscript is clear, well-written, and well-organized. Figures are clear and appealing.

The introduction clearly states the aim of the work and sharply inserts the the work within a well-focused scientific technological framework of general interest.

The theoretical modeling calculations are clear, reliable and strongly founded. The experimental approaches and methods are complete and reliable.

The results are generally interesting. The combination of the theoretical and experimental results allows to elucidate the basic involved parameters and parameters. The discussion of the experimental data is based on reliable general concepts allowing to draw some new and interesting insights.

Overall, i find an interesting manuscript which could be useful for researchers working  in the related fields.

I suggest to the authors only some minor revisions to clarify/improve some aspects:

1) Equation 11: please, justify, from an operational point of view, this assumption. Comment on the basic hypothesis and validity of this approach.

2) Generally, comment on the basic hypothesis of the model calculations by highlighting possible limitations and sources of errors in particular with reference to the experimental conditions.

3) Please, focus more sharply on the concept of roughness by reporting the mathematical definition (surface RMS?)

Author Response

Response to Reviewer 1 Comments

The authors would like to thank the reviewers for the suggestions and valuable comments, which help to improve the quality of the paper. The article was revised according to the remarks.

  • Equation 11: please, justify, from an operational point of view, this assumption. Comment on the basic hypothesis and validity of this approach.

Author: the sentences were added in page 5 line 172:

“It is defined that S = s(x), TIP = tip(u), u0 ≤ u ≤ u1, and S´= s´(x). u0 and u1 are the lower and upper limits of abscissa values of the tip outline separately. There is

When the artefact surface other than the sampling position touches the tip, or in other words, the tip surface other than the tip end touches the artefact surface, the tip-sample convolution occurs. “

  • Generally, comment on the basic hypothesis of the model calculations by highlighting possible limitations and sources of errors in particular with reference to the experimental conditions.

Author: the sentences were added in page 9 line 295:

“Whether the XY stage can move steadily at the defined traverse speed influences the test results. The moving speeds of the XY piezo stage were measured before the ‘flight test’ with laser interferometers and the result proves that the XY piezo stage moves steadily at speeds up to 10 mm/s.”

  • Please, focus more sharply on the concept of roughness by reporting the mathematical definition (surface RMS?)

Author: the sentences were added in page 1 line 25.

“The international standard ISO 4287/4288 [3, 4] specifies various parameters, Ra, Rq, Rz, Rsk, etc., for the evaluation of the surface roughness. The arithmetical mean deviation of the surface height Ra is expressed mathematically by

                                       (1)

where N is the number of measured points in a sampling length, and Zi are the ordinate values of the roughness profile.”

Reviewer 2 Report

The aim of this paper is to investigate the performance of two micro-probes to track sharp surface chances of the structure. The authors have provided a simple model which somewhat helps to simulate the probe in the free-falling state and experimental measurements on selected surface variations. The paper lacks scientific novelty and current organization of the content makes the reading difficult. There is a missing overview of previous research on the topic. The results do not provide more generalized conclusions or new insights about the technique. Therefore my conclusion is rejection.

Additional comments:

Introduction:

What is the range of roughness of interest?

What are the advantages of contact stylus measurement method among other roughness measurement methods? Why this type of micro-probe is used? 

What research has been done on stylus measurement technique? Justify your proposed research. 

What does it mean "The non-linearity between the output voltage variation and the deflection is about 0.3%."? How important is this here?

Not clear what factors influencing the measurement are investigated in the given list 1-4?

Not a clear statement "it can be considered that the dynamics of the cantilever is high enough"?

What does it mean "The measurement bandwidth of the probe"?

State clearly the aim and novelty of the paper.

Modelling:

Separate modeling theory and the results into separate sections with meaningful titles.

What are the assumptions made in eq (1)? It assumes that the cantilever is a rigid body? How did you model the stiffness of the clamped end boundary? What about bending of the cantilever? At least you consider bending vibration in equation (14)? 

How friction force is modeled at an inclined surface?    

Eq (11) is not clear, what physical quantity is surface?

Experimental results:

Separate experimental procedures and the results into separate sections with meaningful titles.

Figure 7: not clear where is the measurement performed, does not coincide with Figure 8 (direction?), where is artefact B? Horizontal scale?

The original depth of the artefact is 10 micrometers, Figure 8 and 9 show something else?

Figure 9 shows that the technique does not measure 90 degree surfaces. Meaning of the result?

What is the largest angle of inclination the method can measure? Does the proposed modeling improve this compared to previous research? 

Author Response

Response to Reviewer 2 Comments

The authors would like to thank the reviewers for the suggestions and valuable comments, which help to improve the quality of the paper. The article was revised according to the remarks.

Introduction:

What is the range of roughness of interest?

Author: the following sentences were added in page 2 Line 66

“The long and sharp tip enables the microprobe for the roughness measurement on the surface with Ra value under 25 µm.”

What are the advantages of contact stylus measurement method among other roughness measurement methods? Why this type of micro-probe is used? 

Author: for the advantages of contact stylus measurement method, following sentences were added in page 1 line 33.

 “The measurement methods of surface texture and roughness can be divided into two categories: contact stylus instruments and optical instruments such as vertical scanning interferometry (VSI). Although the optical methods have the advantages of high throughput and no probing force, their utilization is limited by the optical properties and surface structures of the artefacts. The optical instruments have difficulties in measuring the surfaces with slopes. Undesirable light reflection and diffraction effects can cause poor signal quality. A study performed by Jaturunruangsri proves that the stylus method is more accurate than the VSI instrument in the roughness measurements for hard materials.”

For the reason of using this type of microprobes, following sentences were added in page 1 line 42.

 “However, one drawback of the contact stylus instruments lies in the low throughput. During the measurement the stylus scans line by line. It is especially time consuming for large range measurements. The measurement throughput of the contact stylus instrument depends on the traverse speed of the motion stages during the measurement. The maximum tracing speeds of the state-of-the-art stylus instruments are in the range of 1-3 mm/s. The work of Arvithe Davinci et. al. indicates that the tracing speed of the stylus profilometer effects the roughness measurement results significantly. When the tracing speed is below 500 µm/s, the roughness measurement results are stable. If the tracing speed is further increased, a sharp variation in the results happens. This points out that an improvement of the trackability of the stylus at high speed is necessary.”

In page 3 line 90

“Compared to the existing conventional styli (with masses of several mg, the 1st free resonant frequency in the order of hundreds of Hz, and a cone angle of either 60° or 90°), the microprobe has much lower mass, higher resonant frequency and a sharper tip. It is suggested that the microprobe has a high potential for high-speed measurements.”

What research has been done on stylus measurement technique? Justify your proposed research. 

Author: for the research has been done on stylus measurement technique, following sentences were added in page 2 line 52

“As to the research on more rapid stylus instruments, Morrison developed a prototype stylus profilometer that can measure with speeds up to 5 mm/s in 1995 [8, 9]. Since then there was very little progress on developing stylus instruments with higher traverse speeds.”

What does it mean "The non-linearity between the output voltage variation and the deflection is about 0.3%."? How important is this here?

Author: The following sentences were added in page 2 line 70.

“The microprobe converts the deflection into a voltage output. The nonlinearity of the conversion influences the measurement accuracy directly.”

Sentences were added in page 2 line 73.

“It means that the error caused by the conversion nonlinearity is less than 0.6 µm in measuring a height of 200 µm.”

Not clear what factors influencing the measurement are investigated in the given list 1-4?

Author: four factors of influencing the measurement are listed. The influences of factor 3 and 4 increases with the traverse speed while the influences of factor 1 and 2 keeps constant. We investigated the factor 3 and 4 while we are interested in the performance of the microprobe at high traverse speed.

The sentences were amended in page 3 line 88

“Among the above factors, the demand on the last two factors, dynamics and the measurement bandwidth of the probe, increases with the traverse speed. Hence these two factors become especially important in high-speed measurements.”

Not a clear statement "it can be considered that the dynamics of the cantilever is high enough"?

Author: the sentences were amended in page 3 line 111.

“If the cantilever tracks all the frequency components passing through the low-pass filter formed by the tip, it can be considered that the cantilever can track the surface with fidelity. In other words, the dynamics of the cantilever is high enough for the measurement.”

What does it mean "The measurement bandwidth of the probe"?

Author: the sentences were amended in page 2 line 86

“The measurement bandwidth of the probe is usually defined by the 1st free resonant frequency of the probe.”

State clearly the aim and novelty of the paper.

Author: following sentences were added in page 3 line 95.

“This paper demonstrates the microprobe as a promising stylus probe candidate for high-speed roughness measurement at 10 mm/s. It is intended to give an uncomplicated method to evaluate the trackability of the microprobe. The analysis results were proved with simple and feasible experiments, and the theoretical analysis and experiment results will indicate the improvement direction of the microprobe for a better performance.”

Modelling:

Separate modeling theory and the results into separate sections with meaningful titles.

Author: the modelling theory was separate into 4 parts:

2.1 Theoretical analysis     (page 3 line 119)

2.2 The effect of tip-sample convolution      (page 4 line 163)

2.3 The dynamic behaviour of the microprobe with a silicon tip (page 5 line 182)

2.4 The dynamic behaviour of the microprobe with a diamond tip (page 6 line 219)

What are the assumptions made in eq (1)? It assumes that the cantilever is a rigid body? How did you model the stiffness of the clamped end boundary? What about bending of the cantilever? At least you consider bending vibration in equation (14)? 

Author: the model was modified. The cantilever is now assumed as a point mass in page 4 line 135. Mass-spring-damper model is adopted to calculate the dynamic behaviour of the microprobe. The bending of the cantilever is not considered.

How friction force is modelled at an inclined surface?    

Author: following sentences were added in page 4 line 156.

“The above analysis calculates the sharpest slope feature that the microprobe can track. If the slope is known with angle γ and the trackability of the microprobe on the slope is evaluated, the effect of friction should be considered. The friction Ff is calculated by:

                                                                       (9)

where μ is the coefficient of friction between the microprobe tip and the artefact surface. “

Eq (11) is not clear, what physical quantity is surface?

Author: following sentences were added in page 5 line 172.

“It is defined that S = s(x), T = tip(u), u0 ≤ u ≤ u1, and S´= s´(x). u0 and u1 are the lower and upper limits of abscissa values of the tip outline separately. There is

When the artefact surface other than the sampling position touches the tip, or in other words, when the tip surface other than the tip end touches the artefact surface, the tip-sample convolution occurs.

Experimental results:

Separate experimental procedures and the results into separate sections with meaningful titles.

Author: the experimental procedures were separated into two parts:

4.1 ‘Tip flight’ test (page 9 line 283)

4.2 Artefacts and the test results (page 10 line 301)

Figure 7: not clear where is the measurement performed, does not coincide with Figure 8 (direction?), where is artefact B? Horizontal scale?

Author: the figure of the artefact A (Fig. 7(a), page 10 line 307) was modified and the direction is now same as the measurement. The measurement position was noted in the figure. The figure of artefact B was added to Fig. 7(b), page 10 line 310. Horizontal scale of the both artefacts were noted.

The original depth of the artefact is 10 micrometers, Figure 8 and 9 show something else?

Author: 10 micrometers are nominal values of the artefacts. The real depth of the artefacts can be deviated from the nominal values because of manufacturing errors. Figure 8 and 9 show the measurement values.

the sentence was added in page 11 line 316.

“The measured heights of the both artefacts deviate significantly from the nominal values given by the manufacturer.”

Figure 9 shows that the technique does not measure 90 degree surfaces. Meaning of the result?

Author: The meaning of the result shown in Figure 9 is in page 11 line 331 “the tip vibrated during the measurements on the artefact surface at the higher speeds of 5 mm/s and 10 mm/s. It means that the resonant response of the microprobe with the diamond tip is not high enough for such high-speed measurements. The bandwidth of the microprobe with the diamond tip will have to be improved by increasing the stiffness or decreasing the mass of the tip.”

What is the largest angle of inclination the method can measure? Does the proposed modelling improve this compared to previous research? 

Author: our research indicates that since the trackability of the microprobe with a silicon tip is enough for the traverse speed of 10 mm/s, the largest inclination the method can measure is dependent on the tip form and the mounting angle. 

Page 3 line 107. “sharpest feature that a probe can measure is restricted by the trackability, the tip form and the mounting angle of the probe.”

Page 11 line 319. “Because of the influence of the mounting tilt angle ( = 15°) and the half opening angle of the tip (20°), the measured slope angle of the artefact B is about 55° instead of 90°.”

Page 11 line 324. “the mounting tilt angle ( = 15°) and the half opening angle of the diamond tip (45°) leads an inclination that the diamond tip can measure, which is no more than 30°”

The modelling result only confirms the good dynamics of the microprobe. Compared to previous research, the advantage of this modelling is in page 3 line 97

“The analysis results can be proved with simple and feasible experiments, and the theoretical analysis and experiment results will indicate the improvement direction of the microprobe for a better performance.” 

Round 2

Reviewer 2 Report

The paper has improved considerably. Small remarks:

Figure 2: not clear where is the coordinate origin. 

Eq 2: indicate the direction of the force, e.g Fz(0) 

Author Response

Figure 2: not clear where is the coordinate origin. 

Author: Figure 2 was amended.

Eq 2: indicate the direction of the force, e.g Fz(0)

Author: Equation 2 in page 4 line 132 was modified.       

Following sentences were added or modified in page 4 line 133

 “where z is the vertical deflection variation of the cantilever,  is the deflection force on the cantilever at position 0, Fz (0) is the vertical component of ”
